# Magneto-Mechanical Approach in Biomedicine: Benefits, Challenges, and Future Perspectives

**DOI:** 10.3390/ijms231911134

**Published:** 2022-09-22

**Authors:** Aleksey A. Nikitin, Anna V. Ivanova, Alevtina S. Semkina, Polina A. Lazareva, Maxim A. Abakumov

**Affiliations:** 1Laboratory of Biomedical Nanomaterials, National University of Science and Technology (MISIS), 119049 Moscow, Russia; 2Department of Medical Nanobiotechnology, N.I. Pirogov Russian National Research Medical University, 117997 Moscow, Russia

**Keywords:** nanotechnology, magnetic nanoparticles, magnetic field, magneto-mechanical actuation, remote control

## Abstract

The magneto-mechanical approach is a powerful technique used in many different applications in biomedicine, including remote control enzyme activity, cell receptors, cancer-selective treatments, mechanically-activated drug releases, etc. This approach is based on the use of a combination of magnetic nanoparticles and external magnetic fields that have led to the movement of such nanoparticles with torques and forces (enough to change the conformation of biomolecules or even break weak chemical bonds). However, despite many theoretical and experimental works on this topic, it is difficult to predict the magneto-mechanical effects in each particular case, while the important results are scattered and often cannot be translated to other experiments. The main reason is that the magneto-mechanical effect is extremely sensitive to changes in any parameter of magnetic nanoparticles and the environment and changes in the parameters of the applied magnetic field. Thus, in this review, we (1) summarize and propose a simplified theoretical explanation of the main factors affecting the efficiency of the magneto-mechanical approach; (2) discuss the nature of the MNP-mediated mechanical forces and their order of magnitude; (3) show some of the main applications of the magneto-mechanical approach in the control over the properties of biological systems.

## 1. Introduction

Recent advances in magnetic nanoparticles (MNPs) designed for various biomedical applications, together with their intrinsic magnetic properties, have made them the perfect tools for various biomedical applications, such as magnetic resonance imaging, cell labeling, drug delivery, magnetic separation, biosensing, theranostics, etc. [1,2,3,4,5]. However, scientists are enthusiastically searching for new approaches to implement specific effects of MNPs placed in magnetic fields (MFs). The magneto-mechanical phenomena are some of the most exciting that take place under conditions of externally-applied MFs [6]. Being exposed to these MFs, MNPs undergo mechanical movements in order to align their magnetic moments to the applied field directions. In turn, the movements of MNPs generate mechanical forces capable of manipulating or actuating biomolecules or even cellular structures located in close proximity to MNPs. The ability to manipulate and remotely actuate living cells (or even individual biomolecules inside them_ is one of the most desirable approaches for the modulation and deep understanding of different physiological processes. Non-invasive nature and deep tissue penetration of MFs (in contrast to light) [7] make magneto-mechanical actuation stand out among other stimulation techniques. Additionally, it provides a much more fine-tuned effect with significantly lower MNP concentrations in comparison with traditional magnetic hyperthermia [8,9,10]. The main outcome of the latter is only a local temperature increase whereas non-thermal magnetic exposure may cause highly-specific and even reversible activation of cellular processes.

It is worth mentioning that the MNP response to external MFs is dependent on many different parameters, including intrinsic MNP properties, such as size, shape, magnetic properties, and external properties of MFs, and media, such as the field amplitude, frequency, geometry, temperature, and media viscosity. Each of them is of great importance for magneto-mechanical approach implementation and has a significant impact on the observed biological effects [11]. Moreover, such a variety of parameters makes it mostly impossible to take into account all of them in one theoretical model. Thus, for now, most of the data on the magneto-mechanical approach are based on experimental works made on different MNPs in different external conditions. The main aim of this work was to summarize and simplify theoretical models of MNP interactions with biomolecules under the application of external MFs and to describe current applications of magneto-mechanics in biomedicine.

## 2. Theoretical Foundations of Magneto-Mechanical MNP-Assisted Processes

Basically, three types of applied MFs can be distinguished in magneto-mechanical actuation: alternating magnetic field (AMF, Figure 1a), rotating magnetic field (RMF, Figure 1(b1,b2), and static gradient magnetic field (SMF, Figure 1c). In addition, the combination of RMF with SMF is used to create micro- or nanorobots with drilling motion (Figure 1d) [12]. In each type of external MF, the MNP magnetic moment vector $\vec{\mu }$ will tend to align with the MF vector $\vec{B}$, and the mechanics of this process will depend on many different parameters, which will be discussed in this section.

### 2.1. Néel and Brownian Relaxation Mechanisms

Before considering the magneto-mechanical effects mediated by an individual MNP in different MFs, it is necessary to focus on the mechanisms of the MNP magnetic moment $\vec{\mu }$ relaxation. Spontaneous magnetization of magneto-anisotropic MNPs, i.e., the spin ordering in the absence of an external MF, occurs in two energetically favorable opposite directions, while the imaginary line passing through these directions is called the easy axis of magnetization (Figure 2a). At the same time, the MNP anisotropy energy can be expressed as:(1)$$\Delta E=KV_{\mathrm{MNP}}\sin^{2}\theta ,$$
where *K* and *V*_MNP_ are the MNP effective anisotropy constant and the MNP core volume, respectively; *θ* is the angle between the MNP easy axis of magnetization and the magnetic moment $\vec{\mu }$. In this and the following equations, all units are given in the SI system.

When an MNP is rigidly fixed (unable to rotate) in some matrix, and the thermal energy *k*_B_*T* is greater than the MNP anisotropy energy barrier (*k*_B_*T* > $KV_{\mathrm{MNP}}$), $\vec{\mu }$ can readily flip between these two energetically favorable directions, while an MNP itself is motionless (Figure 2b). This process is called the Néel relaxation, and the relaxation time in the zero field is given by:(2)$$\tau_{N}\sim \tau_{0}exp\left(\frac{\Delta E}{k_{B}T}\right)=\tau_{0}exp\left(\frac{KV_{MNP}}{k_{B}T}\right),$$
where *k*_B_ = 1.38 × 10^−23^ J K^−1^ is the Boltzmann’s constant, *τ*_0_ ~ 10^−9^ s is the attempt time for changes in the dipole direction (the characteristic of an MNP), *T* is the absolute temperature.

Obviously, from the point of view of magneto-mechanics, this type of relaxation is undesirable. When *KV*_MNP_ >> *k*_B_*T* and an MNP is still firmly fixed, then $\vec{\mu }$ is locked in a specific direction of the easy axis and cannot flip between two energetically favorable states (Figure 2c). However, if an MNP is suspended in a fluid, and is able to overcome the energy of the viscous resistance of this medium, then $\vec{\mu }$ rotates together with the entire MNP (Figure 2d). This process is called the Brownian relaxation, and the relaxation time in the zero field is given by:(3)$$\tau_{B}\sim \frac{3\eta V_{\mathrm{HD}}}{k_{B}T},$$
where *η* is the dynamic viscosity of the fluid and *V*_HD_ is the hydrodynamic volume of an MNP.

Equations (2) and (3) indicate that the Néel relaxation in a zero field depends mostly on the internal MNP properties, such as morphology and anisotropy, while the Brownian relaxation in the zero field is also affected by the internal MNP property, such as the hydrodynamic size, but is also affected by the external property, such as the temperature-dependent fluid viscosity *η*. The main relaxation route of $\vec{\mu }$ is determined by the ratio of the characteristic times: for *τ*_N_/*τ*_B_ << 1 and *τ*_N_/*τ*_B_ >>1, then one speaks of the predominance of the Néel and Brownian relaxation, respectively. Generally, for a suspended MNP, both relaxation mechanisms exist, which give the effective relaxation time (*τ*) determined as:(4)$$\tau =\frac{\tau_{N}+\tau_{B}}{\tau_{N}\tau_{B}},$$

It is also important to emphasize that Equations (2) and (3) do not take into account the interparticle interactions and only apply to spherical single-domain MNPs. Any size and shape distributions of MNPs, magnetic dipole interactions, and aggregation will introduce complex deviations in calculations, making these equations practically inapplicable. Moreover, Equations (2) and (3) are applicable only in very small MFs [13]. If an MNP is placed in an external MF, it acquires additional energy ∆*E*_Z_ (Zeeman energy), which lowers the energy barrier of the MNP magnetic anisotropy (Figure 3):(5)$$\Delta E_{Z}=-\mu B=-MV_{\mathrm{MNP}}B,$$
where *M* is the magnetic field *B*-dependent magnetization of an MNP and *V*_MNP_ is the MNP core volume.

Then, Equation (2) will read as:(6)$$\tau_{N}\sim \tau_{0}exp\left(\frac{\Delta E\prime }{k_{B}T}\right)=\tau_{0}exp\left(\frac{KV_{MNP}-MV_{MNP}B}{k_{B}T}\right),$$

Thus, according to Equation (6), *τ*_N_ can vary by many orders of magnitude at the most commonly used amplitudes *B* = 0–100 mT in magneto-mechanical approaches. Even if the MNP magnetic moment $\vec{\mu }$ follows the external MF according to the Brownian relaxation mechanism (*τ*_B_ << *τ*_N_), then with an increase in the field amplitude, the energy ΔE’ value will gradually decrease, which will eventually lead to the Néel regime becoming dominant (Figure 3). Of note, as a rule, MNPs are in the form of cluster-like structures, chains, rings, or aggregates, where the local magnetic dipole interactions of MNPs will also reduce ∆*E*_Z_ by the value *µB*_dip_ and affect *τ*_N_ and the rotational MNP dynamic [14,15].

Since only the Brownian relaxation is of interest for magneto-mechanics, it is preferable to use magnetically blocked highly anisotropic MNPs with large *KV*_MNP_. For each type of magnetic material, there is a critical size of the MNP core (*D*_C_), which corresponds to the crossover between the Néel and Brownian regimes. However, the experimental finding of *D*_C_ is not a simple task. Firstly, this is because the anisotropy constant *K* is a product of both MNP material properties (magnetocrystalline anisotropy) and the MNP morphology (shape anisotropy). Moreover, in contrast to theoretical simulations, in experimentally synthesized MNPs, there is always a certain polydispersity, which undoubtedly causes a wide scatter in *K* values. Secondly, the crossover between the Néel and Brownian regimes also depends on the temperature, the fluid viscosity, and the magnitude of an external MF. It was shown that (in most cases) for iron oxide, MNPs have *D*_C_ ~ 20 nm, while Co-doped iron oxide MNPs have lower *D*_C_ ~ 10 nm due to high values of magnetocrystalline anisotropy [16,17,18,19,20]. It is also very important to keep in mind that the medium viscosity of real systems, e.g., the intracellular environment and various cell compartments, can reach 140 cP [21,22], which leads to a slowdown in the rotational dynamics of MNPs [23]. Under such conditions, aggregated MNPs having a large hydrodynamic size cannot physically rotate. However, if their surface will be protected, for example, with a SiO_2_ shell to prevent the magnetic dipole interactions, Brownian relaxation becomes possible again [24]. Once more, it should be noted that the Néel–Brownian relaxation mechanisms have a lot of limitations, which were mentioned above. However, this is quite enough to understand the fundamentals of the magneto-mechanical approach in external MFs, without going further into the analysis of special cases. Detailed cases of restrictions and extensions to other objects were carried out in various works [25,26,27,28].

### 2.2. Response of the MNP Magnetization to Applied MFs

We begin our consideration with the first and most complicated type of magneto-mechanical MNP-assisted processes, which are supported by AMFs. These types of implementations use electromagnets, and the device designs are essentially the same as in the case of devices for magnetic hyperthermia treatment (MHT) [29]. MHT uses high-frequency AMFs with *f* > 100 kHz, which are considered to be heating fields. Unlike MHT, magneto-mechanics operates with non-heating low-frequency AMFs (LF AMF) with *f* < 10 kHz and *B* << 1 [6,30]. However, in practice, ultra-low frequency fields are most often used, i.e., when *f* < 100 Hz [24,31,32,33,34,35]. At such low frequencies, both the bulk heating of the medium at any MNP concentrations and the local overheating Δ*T*_S_ on the MNP surface can be neglected. One of the main features of magneto-mechanics in AMFs is the dynamic response of an MNP to magnetization. Of all the theories that are devoted to this issue, we will focus on the commonly used linear response theory (LRT) [36,37]. Based on the LRT, the MNP time-dependent magnetization *M* responds linearly with the applied field as:(7)$$M=\tilde{\chi }B,$$
where $\tilde{\chi }$ is the complex magnetic susceptibility of an MNP, defined as:(8)$$\tilde{\chi }\left(\omega \right)=\frac{\chi_{0}}{1+i\omega \tau },$$
where ω is the angular frequency of an MNP, τ is the effective relaxation time, and $\chi_{0}$ is the static magnetic susceptibility of an MNP.

For a randomly oriented MNP ($KV_{MNP}\ll k_{B}T)$, the static susceptibility is described by the Langevin function:(9)$$\chi_{0\;Langevin}=\frac{\mu_{0}M^{2}V_{MNP}}{3k_{B}T},\;$$
while for an MNP with easy axis oriented to the applied field $\vec{B}$ ($KV_{MNP}\gg k_{B}T)$, the susceptibility can be found from the slope of the tangent function [26]:(10)$$\chi_{0\;tan}=\frac{\mu_{0}M^{2}V_{MNP}}{k_{B}T},$$

From Equation (8), it is obvious that the modulus of $\tilde{\chi }$ is calculated as:(11)$$\left|\tilde{\chi }\right|=\frac{\chi_{0}}{\sqrt{1+{\left(\omega \tau \right)}^{2}}},$$

If the external MF follows, for example, a sinusoidal dependence with the maximum amplitude, *B*_0_, and the angular frequency, ω, then the MNP magnetization in such a field also obeys a sinusoidal dependence as (Figure 4a):(12)$$M\left(t\right)=\left|\tilde{\chi }\right|B_{0}sin\left(\omega t+\varphi \right),$$
where φ is the phase lag between the MNP magnetic moment $\vec{\mu }$ and the applied field $\vec{B}$.

From Figure 4b, it is easy to see that the phase lag *φ* can be found as:(13)$$\varphi =arctan\frac{\chi^{\prime \prime }}{\chi^{\prime }}=arctan\left(\omega \tau \right),$$

Next, let us only consider the Brownian regime, i.e., the magnetic moment $\vec{\mu }$ of an individual non-interacting MNP follows the LF AMF vector $\vec{B}$ via the mechanical rotation of the entire MNP, which is suspended in a fluid. Then $\tau =\tau_{B}$, and Equation (13) is transformed into:(14)$$\varphi =arctan\left(\omega \tau_{B}\right),$$

In the ideal case, the maximum of $\chi^{\prime \prime }$ coincides with $\chi^{\prime }$ at *ωτ*_B_ = 1 (or *ω* = *τ*_B_^−1^), which is a characteristic of a single MNP; however, if the dipolar coupling of the MNP is large (chain, rings), then *ωτ*_B_ < 1 [38]. In this regard, for each type of MNP, a critical frequency *ω*_C_ is introduced. If the LF AMF angular frequency *ω* << *ω*_C_, then $\chi^{\prime }\to \chi_{0}$, $\chi^{\prime \prime }\to 0$ and φ
*→* 0 and $\vec{\mu }$ closely follow $\vec{B}$ (Figure 4c). As *ω* increases, the real part of the magnetic susceptibility $\chi^{\prime }$ decreases, while the imaginary part $\chi^{\prime \prime }$ increases, and, as a consequence, the phase lag *φ* also increases. Finally, when *ω* >> *ω*_C_, then $\chi^{\prime }\to 0$, $\chi^{\prime \prime }\to 0$ and an MNP goes into the so-called “frozen state”, i.e., oscillates with extremely small amplitude.

However, in practice, most often, the effective magnetization of the suspended MNP is a superposition of two components: *M* = *M*_N_ + *M*_B_, where *M*_N_ and *M*_B_ are the magnetizations depending on the Néel and Brownian relaxation processes. Deissler et al. showed that for 20 nm magnetite (Fe_3_O_4_) MNPs magnetized in an AMF with various frequencies that the MNP magnetization due to the Néel relaxation mechanism is lagging that of the Brownian even at *f* = 10 kHz and *B* = 100 mT [39]. Thus, before the Néel relaxation mechanism can contribute to effective magnetization, an MNP rotates nearly 180°. Ota and Takemura also showed that the magnetization response of Fe_3_O_4_ MNPs (*D*_MNP_ = 11 ± 3 nm, *D*_HD_ = 44 nm, PDI = 0.13) dominantly occurred by the Brownian regime (*M*_B_/*M*_N_ > 1), when the AMF frequency was *f* < 10 kHz [40]. They also proposed that, in a fluid system, the magnetization in the Néel regime was predominantly affected by the magnetic dipole interactions of MNPs, which reduced *K*, in contrast to the relatively unaffected magnetization in the Brownian regime. Nguyen et al. [41] showed that the transition from the Néel to the Brownian regime did not occur in a continuous way, but abruptly changed around the critical anisotropy constant (*K*_C_), which depended on the AMF frequency and the viscosity of the ferrofluid as:(15)$$K_{C}\left(f\right)=A\left(1-e^{-B\times \left(f+f_{0}\right)}\right),$$
where, A = 214.63 kJ·m^−3^, B = 81.4 ns, and *f*_0_ = 81.27 kHz are fitting constants.

If $\vec{\mu }$ and $\vec{B}$ are not collinear (*φ* ≠ 2πn, n = 0, ½, 1, …), then $\vec{\mu }$ is affected by a magnetic torque $\vec{L}$*,* which can be defined as:(16)$$\vec{L}=\vec{\mu }\times \vec{B},$$

According to Equations (12) and (16), in a sinusoidal AMF, the modulus of $\vec{L}$ changes periodically as:(17)$$L=\left|\chi \right|V_{MNP}B_{0}\sin\left(\omega t+\varphi \right),$$

Obviously, the magnetic torque magnitude will be in the range of 0 ≤ *L* ≤ $\mu B_{0}$. It can be roughly estimated that the spherical Fe_3_O_4_ MNP with *D*_MNP_ = 10–20 nm (*M*_S_ = 92 A·m^2^·kg^−1^, *ρ* = 5200 kg∙m^−3^) in the LF AMF with *B* < 1 T, *L*_max_ ~ 10^1^–10^2^ pN·nm. Remarkably, at first glance, such a small value of a magnetic torque is sufficient for the successful manipulation of various types of macromolecules. For instance, previously using the freely-orbiting magnetic tweezers, the threshold torque leading to dsDNA unwinding (denaturation) was estimated as equal to 10 pN·nm [42,43,44,45,46]. However, in the case of aggregated MNPs, the magnetic moment of each MNP may not be aligned in the direction of the total magnetic moment of the aggregate, which will lead to a lower total magnetization and reduced *L*. Due to the rotational–vibrational movements of an MNP, a magnetic torque is counteracted by a viscous torque from the surrounding fluid, which can be found from Stokes’ law as:(18)$$T=6\eta V_{HD}\omega =\pi \eta D_{HD}^{3}\omega ,$$
where *V*_HD_ and *D*_HD_ are the hydrodynamic volumes and the hydrodynamic diameter of an MNP, respectively; *η* is the fluid viscosity.

In the steady state mode of rotational–vibrational movements, a magnetic torque is equal to a viscous torque, and one can have:(19)$$\frac{\chi_{0}}{\sqrt{1+{\left(\omega \tau_{B}\right)}^{2}}}V_{MNP}B_{0}\sin\left(\omega t+\varphi \right)=\pi \eta D_{HD}^{3}\omega ,$$

As applied to real conditions (*B*_0_ = 10–100 mT, *η ≈* 10^0^–10^2^ cP), *ω*_C_ lies in the range of 10^1^–10^2^ Hz. It is important to note that if there is a binding between two MNPs or an MNP and a surface, e.g., the cell membrane, and this binding is realized by a flexible macromolecular linker or via several macromolecules, then the value of the phase lag *φ* increases as a result of the elastic response of such macromolecules. In some cases, the rigidity of a macromolecular linker will become so high that an MNP cannot oscillate/rotate freely, i.e., the Brownian relaxation mechanism will not be implemented. Thus, two main problems of magneto-mechanics follow from this. Firstly, to correctly calculate the phase lag *φ*, and, as a consequence, the value of the arising MNP-mediated magnetic torque and mechanical forces, it is necessary to know exactly how many macromolecules are bound to an MNP, their elastic properties, the viscosity of the fluid, etc. This is especially important when moderate AMFs in the kilohertz range and MNPs, which are characterized by both relaxation mechanisms, are used. For MNPs with *KV*_MNP_ ~ *k*_B_*T* that cannot resist the elastic response from macromolecules, the transition to the Néel relaxation will occur with a high probability. As a result, the characteristics of an external MF begins to play a key role. For example, Banchelli showed that the clusters of gold-decorated cobalt ferrite (CoFe_2_O_4_) NPs with magnetic core diameters of 6 nm induced DNA melting due to a local temperature increase when exposed to a LF AMF with *f* = 6 kHz and *B* ~ 0.3 T [30]. Unfortunately, the authors did not provide data on the magnetic characteristics of MNPs, which did not allow distinguishing between the contributions from the Néel and Brownian components. On the contrary, theoretical estimations show that the local temperature increase in this case should be insignificant (10^−6^–10^−5^ K) [32,47], which, introduces certain contradictions in the true mechanisms of the observed effects in either case. Secondly, often conclusions about the absolute values of magneto-mechanical forces and torques are made on the basis of experiments with aggregates of MNPs. After that, the resulting value is divided by a factor that takes into account the number of particles in the aggregate, in order to obtain the effect from an individual MNP. We see the main problem with this approach, which is related to the fact that it does not take into account magnetic dipole interactions that affect the MNP anisotropy constant, the magnetic response of particle magnetization to the applied MF, etc.

Unlike periodical oscillations of the MF vector $\vec{B}$ in a LF AMF, a homogeneous AC RMF (Figure 1(b1)) rotates with a constant magnitude and in a two-coordinate plane:(20)$$\vec{B}=[B_{0}\cos\left(\omega t\right),\;B_{0}\sin\left(\omega t\right),0],$$

If $\mu B\ll k_{B}T,$ the MNP dynamic response in an AC RMF essentially does not differ from that in a LF AMF [48]. However, when $\mu B\gg k_{B}T$ and the Néel relaxation becomes a more advantageous mechanism, the above models should be optimized [49]. In the case of uniform RMF implemented using permanent magnets (Figure 1(b2)), it is not necessary to take into account the frequency-dependent dynamic magnetic response of the MNP magnetization. Then, in the steady MNP rotation mode:(21)$$MV_{MNP}B_{0}\sin\varphi =\pi \eta D_{HD}^{3}\omega ,$$
(22)$$\varphi =\arcsin\left(\frac{\pi \eta D_{HD}^{3}\omega }{MV_{MNP}B_{0}}\right),$$

However, the applicability of Equation (22) is limited by the MNP size. Obviously, this size should be such that MNPs do not reorient themselves in an MF under the influence of the Brownian motion. Finally, the magneto-mechanical actuation in SMF with a field gradient ∇$\vec{B}$ essentially does not differ from that realized in the well-known conventional magnetic tweezers technique [50]. In this case, the MNP will additionally experience the pulling force, which will lead to its linear displacements. The same conclusions are also valid for non-uniforms LF AMFs and RMFs.

### 2.3. Deformations and Forces Exerted on Macromolecules Attached to MNPs

If an external non-heating MF acts on MNPs bound to macromolecules, then the MNP movements can lead to the deformations of such macromolecules due to the rising forces of various natures. Obviously, the detectable upper limit of such forces is the binding strength of an MNP to an object, while the lower limit is the thermal motion of an MNP, which induces undirected mechanical vibrations (*k*_B_*T* ~ 4 × 10^−21^ J = 4 pN·nm at *T* = 300 K). All forces arising in magneto-mechanical experiments can be divided into three types: contact, elastic, and hydrodynamic. The macromolecules conjugated with MNPs will experience mechanical deformations when exposed to external MFs. For example, if a flexible macromolecule is anchored between two MNPs or one of its ends is fixed on the MNP surface, while the other end is attached to a rigid substrate (Figure 5a–c), the MNP-mediated contact forces exerted on macromolecular bonds at attachment points in uniform oscillating/rotating MFs can be found as:(23)$$\left|\vec{F}\right|=\frac{\left|\vec{L}\right|}{R},$$

In Equation (23), *R* is the “lever”; that is, the distance from the center of a magnetic core to an attachment point of a macromolecule. In the absence of an additional organic/inorganic protective shell on the MNP, it can be assumed that *R* ~ *R*_MNP_; however, if a targeted macromolecule is anchored to the MNP surface via an additional organic linker, then *R = R*_HD_. The MNP-mediated contact forces *F*_T_, *F*_C_, and *F*_Sh_ in uniform MFs are comparable and are within the same order of magnitude. Numerical estimations show that their values are in the wide range of 10^−1^–10^2^ pN [6,31,51], which are enough, for example, to disrupt the antibody–antigen interaction or destroy the link between the lipid membrane and the membrane protein [52].

In turn, a magnetic torque *L* applied to the macromolecules is balanced by the restoring forces arising from their elasticity. Based on the known values of a magnetic torque and an elastic torque *δ* of a macromolecule, the real deflection angle *β* of an MNP can be found as:(24)$$\arcsin\left(\beta \right)=L\cdot \delta^{-1},$$

The elastic torque *δ* is a product of the macromolecule rotational stiffness *σ* and the macromolecule attachment area to an MNP. According to the elastic equation, the macromolecule stiffness *σ* can be found as [53]:(25)$$\sigma =\frac{ES}{l},$$
where *E* is Young’s modulus, S and *l* are the length and the area of the deformed element, respectively.

Young’s modulus varies over a wide range of values from several MPa to several GPa [54]. For instance, the Young’s modulus of the tumor cells is *E* ~ 1 kPa [55]. Fibrin fibers have *E* = 1–10 MPa, while for the DNA molecules, *E* was determined to be *E* = 0.3–1 GPa [54], similar to the hard plastic [56]. Thus, for example, for 10–20 nm Fe_3_O_4_ MNP (*M*_S_ = 92 A⋅m^2^⋅kg^−1^, *ρ* = 5200 kg⋅m^−3^) attached to the cell membrane (*E* ~ 1 kPa) and treated by the external sinusoidal LF AMF with *B* = 0.1 T, it is easy to calculate that the maximum MNP deflection angle *β* = 10–20°, which corresponds to a lateral displacement of the MNP edges *R*_MNP_·sin(*β*) ~ 2 nm. When the MNP is exposed to the SMF with a gradient $\nabla \vec{B}$, it experiences the pulling force (Figure 5d) that can be found as:(26)$$F_{G}=\left(\mu \cdot \nabla \right)B,$$

Equation (26) can be reduced to the one-dimensional case:(27)$$F_{\mathrm{Gi}}=V_{\mathrm{MNP}}\chi_{0}H\frac{dB}{di},$$
where *χ*_0_ is the volumetric magnetic susceptibility of MNPs calculated from Equations (9) and (10), *H* is the MF strength, *i* ≡ *x, y, z* is the distance from the magnet.

If a macromolecule is bound to a freely rotating single MNP, it experiences a hydrodynamic stretching force (Figure 5e), which can be quantified using the Stokes equation:(28)$$F_{\mathrm{HD}}=6\pi \eta R_{\mathrm{mol}}\nu ,$$
where *R*_mol_ is the hydrodynamic radius of the globule macromolecule anchored to an MNP, *ν* is the linear velocity of the movement of the center of the globule, which can be expressed as a product of the LF AMF instant angular frequency *ω* = 2π*f* and the MNP hydrodynamic radius *R*_HD_.

The majority of enzyme protein molecules have globular diameters of approximately 3–7 nm [57]. If we consider the AMFs with *f* ≤ 1 kHz, and dilute aqueous solutions (*η* ~ 1 cP) and MNPs with *R*_HD_ = 20 nm, the hydrodynamic force values are very weak and lie in the range of approximately 10^−3^–10^−2^ pN. Of note, all of the above examples are ideal cases. In practice, the quantitative values of *F* are influenced by many parameters, such as the aggregation of MNPs, magnetic dipole interactions, the initial orientation of the MNP magnetic moment vector relative to the MF vector, etc. When two MNPs are placed in a uniform MF and approach within a few nanometers, they also experience the attractive force ${\vec{F}}_{1-2}$ (Figure 5f), which arises as a result of the MNP magnetic dipole interactions [58,59]:(29)$$F_{1-2}\propto \frac{\mu_{1}\mu_{2}}{d^{3}}(1-\cos^{2}\gamma )$$
where *µ*_1_ and *µ*_2_ are the moduli of magnetic moments of the MNP-1 and the MNP-2, respectively, *d* is the distance between the MNP-1 and the MNP-2 and γ is the angle between the $\vec{B}$ and the centerline of two MNPs. It is easy to find that the magnitude of such forces is several times less than the magnitude of the contact forces and is about 10^−15^ N for MNPs with *R*_MNP_ = 10 nm at *d* ~ 10 nm. Table 1 summarizes the data on MNP characteristics and MF parameters, as well as the magneto-mechanical forces implemented in each case.

## 3. Practical Applications of Magneto-Mechanical Actuations

Based on the described mechanisms of the MNP movements in the external applied MFs, three subsections of magneto-mechanical actuation implemented in practice can be distinguished: diffusion-associated magneto-mechanical effects, molecule deformation-assisted magneto-mechanical effects, and supramolecular structure disruptions (Figure 6). Let us take a closer look at each of them.

### 3.1. Diffusion Associated Magneto-Mechanical Effects

As discussed above, MNP, being exposed to oscillating/rotating MFs, can perform rotational–vibrational movements, which can lead to an increase in the speed of the local motion of the solvent molecules and macromolecules attached to the MNP surface. In both cases, this increase can be represented as an increase in the local diffusion coefficients of such molecules, leading to changes in diffusion-dependent properties, for example, changes in enzyme catalytic activity. This approach can effectively be used to increase the rate of substrate migration to the active sites.

For example, the effect of a LF AMF-exposed Fe_3_O_4_ MNP with immobilized enzyme laccase (MNPs-laccase) on the rate of the catalytic enzymatic reaction was investigated at different frequencies *f =* 50–600 Hz and amplitudes *B* = 0.4–2 mT [68]. *f* = 600 Hz and *B* = 1 mT resulted in an increase in the MNP-catalyzed reaction rate about 2.1 times compared to the control without any stirring. Notably, the mechanical stirring at 150 rpm led to an increase in the reaction rate of only 1.16 compared to the control without any stirring. Te MNP-assisted increase in the reaction rate in a LF AMF is explained by the increased mobility of the laccases molecules. Fixed on the MNP surface, such molecules tend to align with the direction of the constantly changing MF. MNP with immobilized laccase behaves similar to a microscopic stirrer under the action of an external LF AMF, facilitating the diffusion of the substrate from the liquid into the immobilized laccase and a product from the immobilized laccase into the liquid. The results also show that the reaction rate catalyzed by MNP-laccase increased 1.14 times with the increasing LF AMF frequency from 50 to 600 Hz (*B* = 1 mT). When *B* increased from 0.4 to 2 mT at *f* = 600 Hz, the reaction rate also increased 1.15 times, which confirms the nanostirring hypothesis.

In another work, the same group developed a new fixed-bed reactor to achieve continuous decomposition of phenolic compounds in a high-gradient MF [69]. They further investigated the different effects between continuous and pulse treatments with AMF and found that the rate of continuous treatment for 18 h resulted in a 2.38 times faster reaction rate than the pulse treatment for 6 cycles. It is worth noting that the degradation rate was maintained at over 70% for 48 h by treating MNP-laccase in a fixed bed reactor, which showed great potential for continuous degradation of phenolic compounds.

Similar work was carried out by Lui et al. [70]. They reported that the cross-linked lipases (CLEAs) immobilized on Fe_3_O_4_ MNPs (MCLEAs) acted as microscopic agitators and were used as biocatalysts to catalyze the separation of (*R*,*S*)-2-octanol in a LF AMF. Moreover, they showed that the enzyme activity was higher under the mechanical stirring at 120 rpm in comparison to no stirring conditions, but lower than in the case of LF AMF exposure. In addition, enantioselectivity under a LF AMF exposure was examined and no significant changes were found, indicating that the mobility of MNPs in a LF AMF does not change the structure of the enzyme, but simply enhances the diffusion of the substrate from the liquid into MCLEAs and the product from MCLEAs into the liquid. This indicates that a LF AMF has no effect on the internal characteristic of the enzymatic activity and supports the hypothesis that the enhanced enzymatic activity of MCLEAs is due to the increased mobility of MNPs in the external applied MF.

Cui et al. developed a three-phase fluidized bed reactor with cellulose immobilized on Fe_3_O_4_ MNPs (MIC) as a biocatalyst for the production of chitooligosaccharides from chitosan [71]. A comparative evaluation of MIC activity in a fluidized bed reactor under different types of MFs (SMF, AMF, and PMF-pulse magnetic field) was performed. MIC activity was higher in the PMF than in both SMF and AMF at the same MF amplitude; enzymatic activity reached its maximum amplitudes at *B* = 6.28 mT. The initial increase in MIC activity was due to the application of MFs that increased the frequency rate of contacts between the MIC and the substrate. However, as the PMF intensity increased further, the magnetic agglomeration occurred as a result of the combination of the increasing MF and the high viscosity of the substrate (11.4 mPa·s) at 20 °C, leading to the decrease in MIC activity. Differences in changes in the enzymatic activity of MIC in a SMF and an AMF also depend on the MF amplitude. When *B* was > 0.63 mT, the MIC activity under the AMF exposure showed a decreasing trend with increasing intensity but was steady under the SMF, which may have been due to the alteration of the conformation of MIC and, therefore, affecting the active site of the enzyme.

Tang et al. investigated the catalytic activity of Candida Antarctica lipase B (CALB) immobilized on Fe_3_O_4_ MNPs in comparison with free CALB [72]. Under the influence of a LF AMF with *f* = 25–500 Hz and *B* = 0.28–16.23 mT, they demonstrated the increase in the reaction rate in the case of CALB immobilized on MNPs, which also supported the key role of nanostirring effects produced by MNPs. Of note, the obtained results were used to increase the catalytic efficiency of biodiesel production. Thus, the magneto-mechanical-assisted enzyme activation demonstrates promising potential in increasing the enzymatic reaction rate as well as enzyme stability, which undoubtedly deserves further attention from the point of view of biomedical applications.

### 3.2. Molecule Deformations

As mentioned in the first section, under the influence of external MFs, MNPs generate stretching, compression, and shear forces, which are transferred to the macromolecules linked to these MNPs. Macromolecules may undergo structural deformation resulting in changes in their specific 3D structures leading to conformational and functional changes.

Efremova et al. studied the effect of a LF AMF on the catalytic activity of α-chymotrypsin (CT), which was immobilized on the surface of gold-coated Fe_3_O_4_ MNPs (Fe_3_O_4_–CT) with a magnetic core diameter of 9 ± 2 nm and overall diameter of 25 ± 3 nm [73]. The enzyme reaction rate was measured in situ during the exposure to a LF AMF.

The Fe_3_O_4_–CT conjugate was subjected to mechanical deformations with the reorientation of MNPs at a frequency of *f* = 16–410 Hz and *B* = 88 mT, which led to conformational changes in the structure of CT and a decrease in the initial activity of the enzyme by 63%.

In another work, the same group used the analogs Fe_3_O_4_-Au MNPs bound to CT in two different ways, and analyzed the effects of mechanical activation of protein globules after exposure to a LF AMF (*B* = 140 mT, *f* = 50 Hz), in a pulsed field mode [66]. The theoretical modeling of the effect of the tensile force generated by the MNP motion on the structure of CT was performed; it was suggested that conformational changes occurred in the substrate-binding site but not in the catalytic site. After the treatment with a LF AMF, the Michaelis–Menten constant increased, but the apparent catalytic constant remained the same. UV-Vis and ATR-FTIR spectroscopy revealed some changes in the secondary structure of the enzyme, but the reversible nature of these changes confirmed the fact of stable catalytic activity after magneto-mechanical treatment. The authors claimed that it was the Brownian relaxation of MNPs that led to the stimulation of the conjugated enzyme molecules by MNP-mediated magneto-mechanical forces.

DNA duplexes are another example of stable 3D structures that can be disrupted by applying magneto-mechanical forces. This unique property allows to magnetically trigger DNA dissociation in a LF AMF and opens up new prospects in the field of nucleic acid-based therapy. In [51], the authors used the individual Fe_3_O_4_ MNPs with *D*_MNP_ = 11 nm as a tool for remote-controlled mechanical dissociation of the complementary strands of short DNA duplexes (18–60 bp). In brief, a set of single-stranded (ss) DNA targets of different lengths were immobilized on a glass support, while the individual MNPs were conjugated with a complementary fluorescently-labeled ssDNA probe. After hybridization on the glass surface, LF-AMF was applied. Subsequent rotational–vibrational movements of MNPs resulted in the appearance of the magnetic torque and the stretching force, which, in turn, caused the deformation and dissociation of DNA complementary strands under mechanical stress. Thus, it was shown that small individual MNPs are able to pull out the DNA complementary strands with a binding energy of 90 kcal·mol^−1^. The effects of DNA duplex dissociation were time- and sequence-dependent and were adjusted by changing the MF parameters. Thus, this approach can be used to find the unknown energy of intermolecular interactions.

Controlled DNA release based on cluster-like Au@CoFe_2_O_4_ (8.1 nm) MNPs was described in the work [30]. The surfaces of the MNPs were functionalized with ss-oligonucleotides, combined with a semi-complementary chain in the solution, this caused clustering. The application of a LF AMF (*f* = 6 kHz, *B* ~ 0.3 T) induced DNA dissociation with the release of the single strand that induced clusterization. The authors claimed that the reason for the observed effect was a local temperature increase; however, LF AMF with such frequencies can also provide some magneto-mechanical forces leading to DNA dissociation.

Enzymes and DNA not only change their structures after the application of external forces, but direct MNP interactions with biomolecules within living cells allow mechanical forces generated by MNPs to stimulate various biological processes at a cellular level. Mechanosensitive ion channel actuation or magnetic ion channel activation (MICA) technology offers unique opportunities to manipulate molecular targets (i.e., ion channels) in a MF over a significant distance [74]. There are two essential components for performing such a technique: highly-specific antibody/aptamer/ligand-decorated MNP and MF-actuating MNP mechanical motion. Molecular targets undergo mechanical forces and eventually activate downstream signaling as a result of mechano-transduction. So-called magnetic nanotherapy was successfully applied for treating Ehrlich carcinoma in mice [75]. Fibronectin (Fn) and integrins were chosen as targets for magneto-mechanical activation by oscillating aptamer-modified gold-coated MNPs in a LF AMF. At the applied sinusoidal MF of *B* = 10 mT and *f* = 50 Hz, possible thermoactivation is almost negligible. The proposed mechanism includes several stages: the Fn connection with the aptamer on the MNP surface and MNP oscillations under the LF AMF exposure with the following pulling deformations of the Fn protein chain resulting in integrin molecule twitching on the cell membrane surface. Altogether, it caused a caspase cascade induction and subsequent processes of cancer cell apoptosis and necrosis. Moreover, mechanical actuation led to a profound immune response since segmented leukocyte infiltration was observed in the necrotic area. In summary, the proposed method seems rather promising in terms of precise non-heating cancer cell removal (“nanoscalpel” technique) without any harmful effects on healthy tissues.

Another approach for cellular signaling pathway magneto-mechanical activation is receptor clustering. Perica et al. [76] used nanoscale artificial antigen-presenting cells (nano-aAPC), which were constructed by coupling a chimeric MHC-Ig dimer (able to bind T cell receptor—TCR) and anti-CD28 antibody (for T cell response modulation) to 50–100 nm paramagnetic dextran-coated MNPs. Incubation of T cells with nano-aAPC in MF with *B* = 0.2 T provoked TCR aggregation, resulting in T cell activation. Interestingly, bit micro-aAPC did not demonstrate the same effect, despite greater mechanical forces, they should transmit in MF. The authors explained such phenomena by mechanism differences, e.g., in case microparticle mechanical pulling movements take place but no clustering of the receptors. The effectiveness of antigen-specific cell magnetic stimulation was tested in a poorly immunogenic B16 melanoma in vivo model [77]. T cell populations activated in vitro by exposure to MF with nano-aAPC were administered in tumor-bearing mice and had strong inhibitions in tumor growth compared to other controls. The results obtained led to a variety of remarkable conclusions. First, it was determined that the process of T cell activation is strongly dependent on membrane spatial organization, and we are able to manage this process through MNP clustering modulation in MF. Secondly, the magneto-mechanical procedure allows stimulating naive T cell populations for profound anti-tumor effects. Because MF can also be used to deliver, transport, and retain MNP-labeled cells [65,78], it is also possible to perform site-specific T cell activation in vivo. Finally, MNP-based agents can be successfully applied for cellular therapy and immunotherapy protocols in particular.

The clustering approach has also worked for epidermal growth factor receptor (EGFR) magneto-mechanical activation in both A431 epidermoid carcinoma and HeLa cells with high and low receptor expressions, correspondingly [79]. In brief, treatment with “magnetic switches” (monoclonal anti-EGFR antibodies carrying superparamagnetic MNP) and subsequent exposure to MF caused strong magnetic dipolar interactions between MNPs, resulting in their clustering. EGFR-targeted MNPs were able to transmit mechanical forces (in the sib-pN range) to the receptor and provoke transphosphorylation and downstream signaling cascades without any natural ligands. Hence, a ligand-independent way of growth factor receptor activation was demonstrated. The Brownian motion of magnetic switches may induce molecular crowding and control/modulate cellular processes and different functional states. Considering that EGFR and other tyrosine kinase receptors of the ErbB/HER family are upregulated in nearly half of all human cancers [80], magnetic switches may be considered safe and powerful vehicles for anti-tumor therapy.

An accurate approach for osteogenesis promotion was implemented through direct actuation of the TREK1 mechanosensitive ion channel [81,82,83]. MNPs were conjugated with RGD peptides or anti-TREK1 antibodies. To detect the MSC stimulation effects by MNP in MF, MNP-labeled human MSCs were surrounded by a confluent layer of fetal chick epiphyseal cells (CECs). Additional MF effects (array of permanent NdFeB magnets, 25 mT, 1 Hz) resulted in a 60–90% increase in total collagen production in CECs. It was demonstrated that TREK1 magnetic stimulation in MSC resulted in migration facilitation as well as cell mineralization. As shown, a combination of biological factors released by MSC and remote mechanical activation of the ion channel ensured simultaneous control of the tissue response and signaling process. Due to the described magnetic stimulation, it is possible to use MSC in vivo for tissue regeneration and healing and treating endogenous cells in a paracrine signaling way [84].

An example of magneto-mechanical stimulation of TRPV4 (Transient Receptor Potential Cation Channel Subfamily V Member 4) was also presented in [85]. The activation system consisted of two separate parts: TRPV4 fusion protein with a His-tag in the extracellular loop and 100 nm MNPs carrying anti-His antibodies to target specific cells and reduce the concentration of MNPs for appropriate stimulation. In vitro experiments demonstrated receptor activation, resulting in significant Ca^2+^ influx under magnetic stimulation of both HEK293T cells and mice cortical neurons after plasmid transfection or virus transduction correspondingly. Notably, higher amplitudes of the LF AMF induced a more profound release of intracellular calcium. Moreover, after mice exposure to MF (200 mT, permanent magnet), MNP-based nanotools unilaterally provoked the rotation and “freezing of gait” behaviors usually observed in Parkinson’s disease patients [86]. Again, the researchers focused on the non-heating way of receptor activation as compared to earlier magnetothermal approaches for TRPV4 activation [87,88]. This point is particularly important for neurostimulation as local overheating can result in harmful heat dissipation in the brain. The need for two injections instead of one (in the case of genetically-encoded MNPs) may seem more complicated, but it negates off-target effects and demonstrates a highly specific non-invasive (unlike optogenetics) technique for neural modulation, new circuit interpretation, and other clinical applications.

### 3.3. Supramolecular Structure Disruptions

Physical disruption of highly-structured normal state biological membranes is likely the most evident possible consequence of tissue interactions with moving MNP. When exposed to external MFs, MNP experiences different movements, such as oscillation and rotation, which generate mechanical forces with magnitude values enough for direct disorientation of lipid bilayers in liposomes or intracellular membranes. It results in the following signaling pathway activations, i.e., apoptosis, necrosis, reactive oxygen species (ROS) production, etc. [89,90].

Liposomes were loaded with Fe_3_O_4_ MNPs and 5(6)-carboxyfluorescein as a model of a hydrophilic drug [91]. After exposure to 20 kHz, 75 µT MF, the liposome bilayer permeability was affected due to MNP oscillations and the release rate increasing. In another study [92], a more complex vehicle (“nanochain”) was used for glioblastoma treatment in mice. These nanochains were constructed by three 10 nm iron oxide MNPs chemically bound to the doxorubicin-loaded 35 nm liposome. The RGD peptide served as a targeted moiety for better accumulation in the tumor region in the brain. Application of external LF AMF (*B* = 5 mT, *f* = 10 kHz) resulted in widespread delivery of the drug and an increase in the survival of the nanochain-treated animals. No local heating effect was observed, so the authors concluded that the release process was not based on hyperthermia. In [93], several magnetoliposomes of different chemical compositions were tested for dye (calcein) release in the presence of LF AMF at *B* = 2.6–201 mT and *f* = 30–3000 Hz. It was demonstrated that the MNP clusterized in a membrane region and oscillated during the LF AMF exposure, which led to a more intensive release. The intensity of this process also depended on liposome composition and field amplitude. Interestingly, some magnetoliposomes with liquid crystalline membranes containing unsaturated lipids or cholesterol were able to undergo small “healings” against MNP-induced deformations.

In [67], the authors showed how a low-frequency RMF induced U87 glioma cell death, treated with zinc-doped MNPs. It is widely known that the addition of some metal elements, such as Zn, Co, and Ni, increases MNP magnetization, providing a more pronounced response to the applied MFs [94]. MNPs were decorated by the epidermal growth factor (EGF) peptide to be specifically internalized into tumor cells via receptor-mediated endocytosis and then transferred to lysosomes. Under RMF exposure (*f* = 15 Hz and *B* = 40 mT), EGF-MNP formed elongated aggregates along the field direction, which were able to rotate. Mechanical forces generated by anisotropic aggregates (hundreds of pN), result in efficient damaging of lysosomes and plasma membranes stimulating both apoptotic and necrotic cell death pathways. The authors also speculated about possible magnetic activation of mechanotransduction channels on the cancer cell membrane, but this statement needs further investigation. Nevertheless, permeability changes and the following physical disruption of cellular membrane structures in low-frequency RMF could be powerful tools for non-heating brain tumor treatment.

EGFR-targeting was also applied in the case of head-and-neck squamous cell carcinoma A431 (HNSCC) cell culture, but the type of MF used for mechanical lysis was different [95]. Briefly, 50, 100, and 200 nm of PEG-coated Fe_3_O_4_ MNPs were decorated by cetuximab (a chimeric mouse–human monoclonal antibody to EGFR) and assessed for their efficacy after an application of the LF AMF (*f* = 4 Hz). In vitro experiments revealed complete human HNSCC cell death. Apparently, MF could generate oscillatory magnetic torques on the MNP, resulting in shear forces able to break the lysosome integrity and provoke its leakage in the cytoplasm, inducing apoptosis [96]. Meanwhile, all types of MNPs significantly decreased the tumor growth rate in vivo after six AMF treatments. The highest antitumor effect was observed for 200 nm MNPs, which is likely explained by the lower clearance rate from the tumor tissue [97]. The necrosis areas were visualized in tumor tissues by MRI supporting the therapeutical effects. The authors especially emphasize that cancer cell death was not caused by local heating (as it takes place in traditional hyperthermia) but rather by a mechanical force produced by MNP.

The idea of using physical and especially magnetic stimuli for cell proliferation was also supported by creating a bioreactor for building new tissue cultures [98]. Magnetically-responsive scaffolds based on CoFe_2_O_4_ MNPs (*D*_MNP_ = 35–55 nm) are key players in the proposed technique, serving as support for cells and mechanical stimulation generated by external MFs, transmitting them to the surrounding cells [99]. For example, to stimulate the MNPs within scaffolds, the magnetic field must reach values from 20 to 50 mT [100]. As a proof of concept, MC3T3-E1 pre-osteoblast cells were used as a model of bone regeneration. The cells were treated with two cycles of magnetic stimulation resembling the human body’s daily mechanical conditions. The bioreactor platform was able to maintain cell culture conditions and biocompatibility, and provide effective cell adhesion, growth, and proliferation. In other words, it allows mimicking the real tissue microenvironment with both biochemical and biophysical signals, and evaluating their impacts on cell behavior. Taking into account that magnetoactive scaffolds can be arranged in a human body, this bioreactor is a highly promising tool for the remote stimulation of cells in patients and other applications in the frame of tissue engineering.

### 3.4. Other Applications

In addition to therapeutic applications, the magneto-mechanical approach can also be used for targeted diagnostic purposes to capture the cellular and molecular processes underlying pathological disorders in a living organism. The ability of an MNP to move being exposed to an external time-varying MF was implemented in a new diagnostic technique called magnetomotive ultrasound (MM US) [101]. MNPs, usually superparamagnetic, serve as contrast agents and have significant advantages over microbubbles traditionally used in US imaging technologies. Due to their sizes, compositions, and surface properties, MNPs can reach extravascular targets, whereas micrometer-sized microbubbles are limited by vascular systems [102]. The MNP, being captured in a certain body compartment, starts moving in the MF generated by permanent and/or electromagnets, and provokes the movement of the surrounding tissues. These movements can be detected with a US using frequency- or time-domain analyses of echo data [103]. This visualization method was successfully applied for the diagnostics of stomach abnormalities [104,105] and sentinel lymph nodes [106,107] in rats. In summary, there is an upgrade of real-time, non-radiation, easy-to-use, and widely available US techniques in terms of highly-specific nanoparticle accumulation, especially in the case of the targeted moiety presence on their surface. It significantly increases the spatial resolution and broadens horizons for precise molecular imaging of pathological processes. Moreover, taking into account that movements of surrounding tissues depend on their stiffness, MMUS technology can be used for tissue mechanical property investigations using MNPs and applied MFs with different characteristics. As for clinical applications, MMUS may become a useful tool for preoperative examination and surgical guidance without such limitations as radioisotope exposure or low penetration depth of light-based imaging systems. However, some necessary technical developments are still needed [103,108]. Nevertheless, the MMUS technique has some limitations, such as a poor signal-to-noise ratio, a comparatively low spatiotemporal resolution, and can be improved by using magneto-gas vesicles (MGVs) as alternative contrast agents [109]. Gas vesicles (GVs) represents air-filled protein nanostructures, which can be easily isolated from photosynthetic microorganisms [110]. Their unique physical properties, especially the significant scattering of sound waves, make them a perfect platform for the MMUS contrast agent design [111]. Being loaded by MNPs, they can serve as magneto-acoustic nanostructures for non-invasive evaluation of tissue mechanical properties and their changes through pathological process developments. For example, zinc-doped iron oxide MNP-based MGVs were effectively used for the MMUS detection of fibrosis in both lung and liver organoids due to increased tissue stiffness. Moreover, these novel contrast agents discriminated between fibrotic and normal liver tissues in mice, displaying their advantages over GVs and MNPs alone. These impressive results significantly broaden the MMUS in vivo applications in quantitative analyses of localized tissue mechanics or diagnostics of diseases accompanied by stiffness changes. Finally, as GVs can be genetically encoded in living cells [112], it is possible to target them to a specific tissue in order to assess local molecular signals and processes that underline disorder progression.

Reproducing the actual physiological mechanistic processes became real due to the multi-component imaging-stimulation framework developed and presented in [113]. The key element of the system is magneto-rheological elastomers [114,115] consisting of polymer matrix and micron-sized magnetic particles and used as a cellular substrate. Particle magnetization in MFs induces mechanical forces, which in turn are transferred to the matrix causing the deformations and subsequent mechanical activation of living cells cultured on the substrate. MF is provided by four sets of mobile permanent magnets around the sample and each one can be controlled independently to provide the necessary magnetic field magnitude and direction. An advanced computational framework directs the controlling parameters to create a certain deformation pattern to provoke in vitro biological responses within the substrate. By a set of computational simulations, the developed system was able to reproduce heterogeneous brain strain distributions during closed-head impacts described in [116]. Additionally, mechanical stimulation was successfully implemented for human dermal fibroblast cellular alignment. In other words, the stimulation system proposed is a very promising approach for modeling different physiological scenarios connected with mechanistic changes, such as fibrosis, traumas, skin scaring, and others.

The magneto-mechanical approach can also serve as a quantitative bioassay for the presence of target molecules (for instance, antigen or DNA) in a solution [117]. The principle of this method is based on the measurement of the phase lag change $\Delta \varphi =\varphi_{2}-\varphi_{1}$ between the MNP magnetic moment $\vec{\mu }$ and the MF vector $\vec{B}$ before and after a functionalized MNP specifically binds to the target molecules producing the antigen–antibody or DNA–DNA interaction. According to Equations (3) and (14), one can see that Δφ strongly depends on the hydrodynamic size of MNP at a constant MF frequency and amplitude. Thus, if the specific binding of an MNP to the targeted molecules occurs, the increased MNP hydrodynamic size results in increased *τ*_B_ and $\varphi_{2}$, which can be easily detected.

Most of the observed effects on living cells have been detected based on biochemical responses, which are not just consequences of cascades of signals that are transformed from mechanical to electrical or biochemical; often, the application of MFs and the detection of the effects are nor recorded simultaneously. Fluorescent microscopy is one of the techniques that allow the simultaneous application and recording of MF; it is a powerful tool used in detecting changes in biochemical reactions in situ due to a high variety of different specific dyes and markers. However fluorescent microscopy is not able to detect direct changes in mechanical properties of biological objects on scales of tens or hundreds of nanometers. Ion conductance microscopy is another powerful technique that allows such measurements [118]. This technique was shown to be able to measure the nanomechanical properties of living cells; moreover, it has no detectable effects on them. Ion conductance microscopy, combined with confocal microscopy, can provide new insight into the mechanism of magneto-mechanical effects on the molecular level.

## 4. Conclusions

Magneto-mechanical actuation is an actively developing area of research (compared to the novel ways of application, which show up massively). Due to a large variety of MFs and MNP parameters, which are critically important for biomolecule manipulations, we attempted to clarify and systemize the physical bases and practical applications of this approach. Below, we highlight some valuable aspects of the current review:The forces experienced by MNP in three different types of applied MFs, which are usually used in the magneto-mechanical technique, may lead to movements of the macromolecules attached to the nanoparticles.The currently known magneto-mechanical-induced biological effects mostly include diffusion changes, biomolecule deformations, and membrane disruptions.Despite the safety of MFs, and the incredible progress in the creation of biocompatible, colloidally stable MNPs, there is a lack of toxicity and possible side effect studies on the magneto-mechanical approach.Possible problems with MNP aggregation and non-specific interactions with biomolecules in a living body may cause translation difficulties from in vitro to in vivo. The creation of novel equipment for MF generation within magneto-mechanical actuation in humans could be another obstacle to scaling up the approach.Highly target-specific MNP designs and syntheses broaden the horizons for the precise control and modulation of various physiological conditions at the cellular and molecular levels, both in vitro and in vivo.

We have a long way to go before finding the correlation between MNP and MF characteristics and the following biological outcomes (despite being promising). We strongly believe that a data-driven approach will allow identifying the common patterns of the phenomena observed. Finally, the design of new magnetic probes as well as the search for alternative molecular targets, together with the replacement of local nanoparticle administration for systemic, are the most obvious challenges that researchers will meet regarding increasing magneto-mechanical actuation specificity and efficiency.

## Figures and Tables

**Figure 1 ijms-23-11134-f001:**
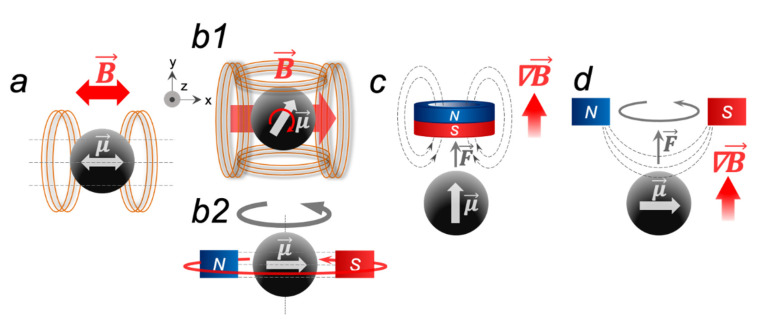
Different routes of an MNP activation by external MFs. In such fields, the magnetic moment $\vec{\mu }$ of an MNP aligns with the field vector $\vec{B}$: (**a**) uniform AMF; (**b1**) RMF implemented by the two-axis perpendicular Helmholtz coils 90°, out of phase with each other; (**b2**) RMF implemented by two permanent magnets that rotate mechanically; (**c**) SMF with the filed gradient ∇$\vec{B}$. (**d**) The combination of (**b2**,**c**).

**Figure 2 ijms-23-11134-f002:**
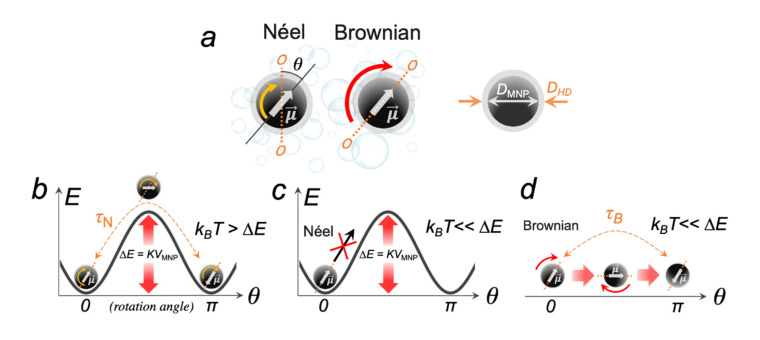
Schematic representation of the dynamic behavior of the MNP magnetic moment $\vec{\mu }$ in the absence of an external MF; (**a**) Néel and Brownian relaxation mechanisms, where *θ* is the angle between the MNP magnetic moment $\vec{\mu }$ and the easy axis of magnetization *O-O*; *D*_MNP_ and *D*_HD_ are the core and the hydrodynamic diameters of an MNP; (**b**) Néel relaxation in a zero field when *k*_B_*T* >> *KV*_MNP_; (**c**) Néel relaxation is forbidden if the MNP is physically unable to move and *k*_B_*T* << *KV*_MNP_; (**d**) Brownian relaxation becomes possible if an MNP is suspended in a fluid.

**Figure 3 ijms-23-11134-f003:**
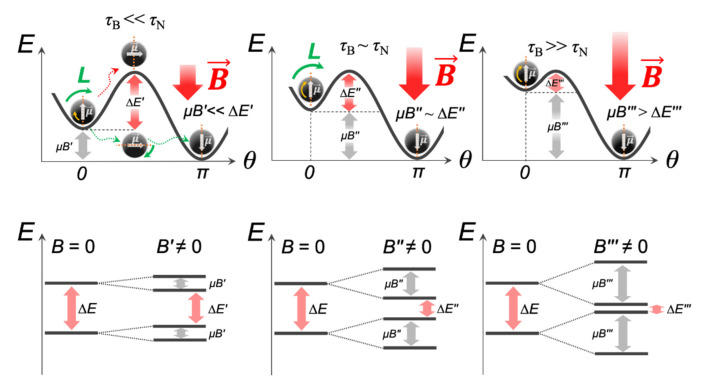
An increase in the amplitude of an external MF leads to a decrease in the value of the anisotropy energy barrier ∆*E′* > ∆*E″* > ∆*E‴* and gradual dominance of the Néel regime. *L* denotes the magnetic torque.

**Figure 4 ijms-23-11134-f004:**
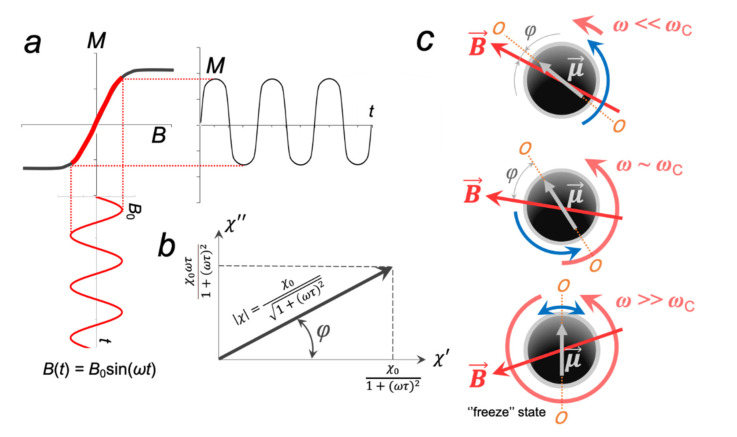
Dynamic response of an MNP to an applied external AMF: (**a**) magnetization response of an MNP to an alternating sinusoidal field *B*; (**b**) frequency-dependent magnetic susceptibility of an MNP, where *τ* is the effective relaxation time, $\chi^{\prime }$ and $\chi^{\prime \prime }$ are the real and the imaginary parts of magnetic susceptibility, respectively; (**c**) dynamic behavior of the MNP magnetic moment $\vec{\mu }$ in AMFs of various frequencies.

**Figure 5 ijms-23-11134-f005:**
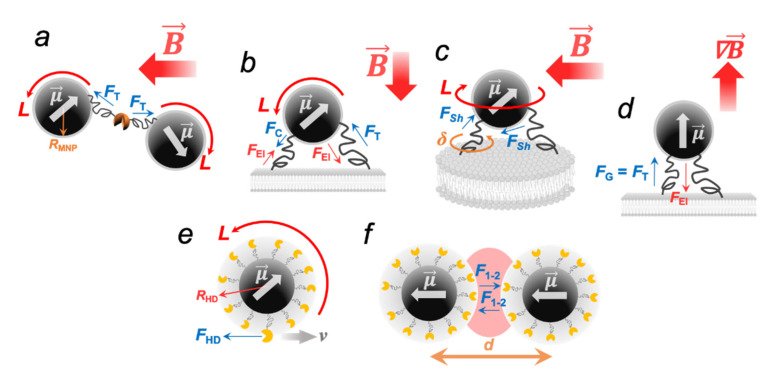
Forces and torsion stress exerted on macromolecules anchored to MNPs: (**a**–**c**) oscillating/rotating MFs; (**d**) after exposure to SMFs with a filed gradient ∇$\vec{B}$; (**e**) hydrodynamic force *F*_HD_ acting on a macromolecule during the MNP rotational movements; (**f**) attractive force *F*_1–2_ arising between two MNPs as a result of magnetic dipole interactions. *F*_T_, *F*_C_, *F*_Sh_ are the contact forces of stretching, compression, and shear, respectively; *F*_G_ is the pulling force generated by the filed gradient (analog to *F*_T_); *δ* is the torsional stress arising on the macromolecules. The elastic force *F*_El_ tends to return a macromolecule to its natural shape.

**Figure 6 ijms-23-11134-f006:**
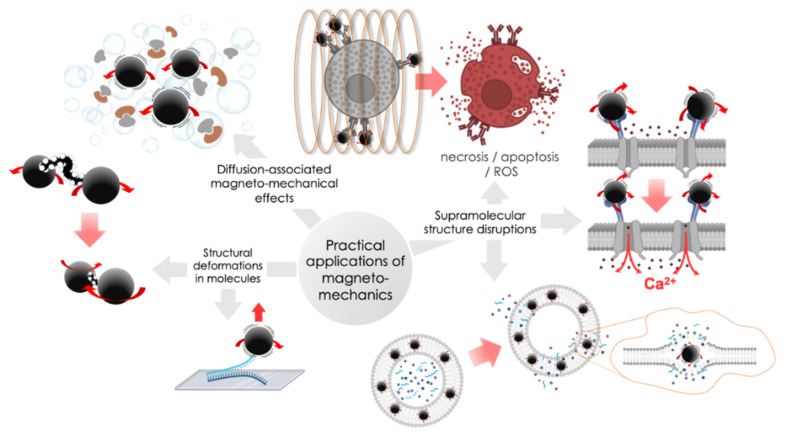
Schematical representation of magneto-mechanical actuations realized in practice.

**Table 1 ijms-23-11134-t001:** MNP-mediated magneto-mechanical effects in various MFs.

Core	Coating	MNP Parameters		MF Parameters			Force, *F*	Reference
		Core Size, *D*_MNP_	Hydrodynamic size, *D*_HD_	Frequency, *f*	Amplitude, *B*	Gradient, ∇*B*		
SMF								
Iron oxide	Carboxydextran	5 nm	30 nm	-	n/a	10^4^ T/m (max)	1–100 pN (*N* = 10^2^–10^3^ MNPs in cluster)	Magnetically driven cellular endocytosis [60]
	Cross-linked polymer hydrophilic polymer with carboxylic groups	~100 nm (Spherical cluster containing SPIONs)	~120 nm	-	150 mT	50 T/m	10 fN per 1 MNP 100–200 pN (N = 4·10^4^)	Spatiotemporal control of microtubule nucleation [61]
	Citrate molecules	5 nm	n/a	-	~100 mT	120 T/m	60 ± 20 nN	Control of drosophila embryonic tissue deformation [62]
Zn-doped iron oxide	SiO_2_ shell (3.8 nm)	50 ± 4 nm	30 nm	-	n/a	10^3^ T/m	0.1 pN per 1 MNP	Mechanical control of the inner earhair via the gating of mechanosensitive ion channels [63]
	DR4 antibodies	15 nm	~120 nm	-	0.5 T	n/a	*F*_G_ (1 MNP): 9.2·10^−^^19^ N *F*_1__–__2_ MNPs: 3.4·10^−^^14^ N	Control of DR4 receptor activity [64]
	Dextran	15 nm	n/a	-	1 T	n/a	1.6–61 pN	In vivo control of stem cell migration and differentiation [65]
**LF AMF**								
Iron oxide	Poly(ethylene glycol)	11 ± 2 nm	28 nm	180 Hz	100 mT	10^−4^ T/m	2.1 ± 0.4 pN	Mechanical dissociation of complementary strands of short DNA duplexes [51]
	Poly(L-lysine)-block-poly(ethylene glycol)@ superoxide dismutase 1	9 nm	~110 nm	50 Hz	150 mT	-	*F*_HD_ ≈ 10^−1^ pN	Remote control of superoxide dismutase 1 catalytic activity immobilized on MNPs [32]
	Poly(ethylene glycol) SiO_2_@Poly(ethylene glycol)	9.5 ± 1.1 nm 23.0 ± 1.8 nm	28 nm 68 nm	19–211 Hz	100 mT	-	5.3 · 10^−^^14^ N (per 1 MNP) 2.1· 10^−^^14^ N (per 1 MNP)	Intracellular membrane integrity failure [24]
	Poly(ethylene glycol) based polymer	8 nm	12 nm	50 Hz	100 mT	-	3 pN	Cancer cell-selective treatment through cytoskeletal disruption [35]
	Au@lipoic acid-α-chymotrypsin Au@cystamine-α-chymotrypsin	Iron oxide: 9 ± 2 nm Iron oxide@Au: 25 ± 3 nm	171 ± 3.9 nm 113 ± 1.6 nm	50 Hz	140 mT	-	80 pN	Remote control of α-chymotrypsinactivity [66]
**RMF**								
Iron oxide	Phosphonate pegylated ligands bearing carboxylate functions@gastrin	6.0 ± 1.3 nm	43 ± 4 nm	1 Hz	30–60 mT 50 mT >100 mT	-	(*N* = 3·10^3^ MNPs in cluster) 1–3 pN 3 pN 0.7 pN	Mechanical activation of magnetic nanoparticles induced lysosome membrane permeabilization and the release of the lysosome content and cell death [34]
	Poly(ethylene glycol) @EGF peptide	62.1 ± 0.8 nm	220 nm	15 Hz	40 mT	0.03 T^2^/m (x-z plane)	~0.008 pN	Programmed cell death and necrosis [67]
20:80% iron–nickel	Gold (5 nm)	Disk-shaped MNPs: 60-nm-thick, ~1-µm-diameter	n/a	<60 Hz	90 Oe	-	10^1^ pN	Compromised integrity of the cellular membrane and initiation of programmed cell death [31]

## Data Availability

Not applicable.

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
