# Peer review of "Magneto-Mechanical Approach in Biomedicine: Benefits, Challenges, and Future Perspectives"

_ijms, 2022, doi:10.3390/ijms231911134_

Round 1

Reviewer 1 Report

1. In accordance with the title of the paper: "Magneto-mechanical approach in biomedicine: benefits, challenges and future perspectives", I suggest that the introduction needs to be updated by including and discussing the advantages and the disadvantages of the magnetochemical approach in biomedicine.

2. I strongly recommend to the authors that the conclusion should be added as a separate section. The conclusion should be more highlighted by future perspectives and general tendency.

Author Response

We thank Reviewer for valuable comments and agree that the “Introduction” and “Conclusion” were poorly organized. We took into account all the suggestions and finalized the manuscript.

1) The "Introduction" has been updated by including and discussing the advantages of the magneto-mechanical approach in biomedicine, whereas the discussion of possible obstacles on a way to clinical implementation was done in Conclusion section.

2) The conclusion has been added as a separate section. Here, we summarize the main features of magneto-mechanics and highlight issues that need to be given special attention.

Reviewer 2 Report

The manuscript "Magneto-mechanical approach in biomedicine: benefits, challenges and future perspectives" by Nikitin et al. is a very-well written review on applications of magnetic technologies in biomedicine.

The manuscript mainly focuses on therapy while biomedicine also can suggest diagnostics (where magnetic approaches are as valuable). Thus, I would suggest to add a phrase why diagnostics/analytical approaches are not included in this review.

I only detected some minor issues such as a few typos or the separation of two brackets such as [108],[109] in the text. I recommend to accept the manuscript with minor revisions.

Author Response

We thank Reviewer for its remark and totally agree with it. We have rewritten some sentences and added the information where the magneto-mechanics can serve as a diagnostic tool (section “3.4 Other applications”). In addition, we would like to note that we deliberately ignored the well-known and well-proven technique of magnetic tweezers, as this technique uses micrometer magnetic particles. The development of analytical techniques based on magneto-mechanical approach is still in very preliminary phase; however, we have found some works that can serve as a promising tool for diagnostic systems creation (lines 721 – 729).

We have also corrected any inaccuracies in the text, such as two separate brackets [108], [109].

Reviewer 3 Report

The manuscript "Magneto-mechanical approach in biomedicine: benefits, challenges and future perspectives." has an actual and interesting subject of the research.

The work presents necessary, interesting and useful aspects, being well conceived and substantiated in the theoretical part.

However, the introduction is too brief and does not make references to some excellent works close to these researches.

The part of applications and subsequent challenges is not as well supported and illustrated (figures, diagrams, images), as the theoretical part was beautifully presented.

Moreover, the title of the work prepared us for more consistent and more strongly supported applications with strong references.

On the other hand, the challenges for the field should be better scored and found including in a consistent paragraph of the conclusions.

Author Response

We are grateful for Reviewer comments and believe that all questions raised are answered in the following text.

1) However, the introduction is too brief and does not make references to some excellent works close to these researches.

We have extended the “Introduction” by including the works close to the topic of this review, and discussing the advantages and disadvantages of the magneto-mechanical approach in biomedicine.

2) The part of applications and subsequent challenges is not as well supported and illustrated (figures, diagrams, images), as the theoretical part was beautifully presented.

We agree with this comment, however we believe that Figure 6 illustrates all possible applications listed in our review and simple splitting into separate figures will not make manuscript more informative.

3) On the other hand, the challenges for the field should be better scored and found including in a consistent paragraph of the conclusions.

The conclusion has been added as a separate section. We summarized the main features of magneto-mechanics and highlighted issues that need to be given special attention.

Reviewer 4 Report

Proposed review is a well-prepared and interesting work. It provides lots of information and is based on numerous references (111 literature reports). The only minor revisions which are suggested concern the following issues:

1) Abstract of the paper should be extended to provide more information on the review.

2) Second paragraph of Introduction needs to be supplemented with references.

3) All parameters included in the equations of the article should be supplemented with units.

4) Notations of the polymer names should be improved, e.g. it should be "poly(ethylene glycol) instead of "polyethylene glycol" (Table 1.).

5) Section "Conclusions" should be added in which Authors will sum up the review and provide highlights of their work.  

6) Section References needs to be corrected to be consistent. Now, some references contain the whole journal names while the other ones contain their abbreviations.

Author Response

We thank Reviewer for valuable comments and hope that all questions have been answered.

1) The “Abstract” have been extended to provide more information on the review.

2) Second paragraph of Introduction was supplemented with references

3) Information that all units are listed in SI system is added to 2.1 section after first equation appears (line 83).

4) Designations of polymer names have been standardized

5) The conclusion has been added as a separate section. Here, we summarize the main features of magneto-mechanics and highlight issues that need to be given special attention.

6) Section “References” has been corrected.
